# Metastatic Prostate Cancer Cells Secrete Methylglyoxal-Derived MG-H1 to Reprogram Human Osteoblasts into a Dedifferentiated, Malignant-like Phenotype: A Possible Novel Player in Prostate Cancer Bone Metastases

**DOI:** 10.3390/ijms221910191

**Published:** 2021-09-22

**Authors:** Cinzia Antognelli, Lorella Marinucci, Roberta Frosini, Lara Macchioni, Vincenzo Nicola Talesa

**Affiliations:** 1Department of Medicine and Surgery, Bioscience and Medical Embryology Division, University of Perugia, L. Severi Square, 06129 Perugia, Italy; lorella.marinucci@unipg.it (L.M.); roberta.frosini@unipg.it (R.F.); vincenzo.talesa@unipg.it (V.N.T.); 2Department of Medicine and Surgery, Biochemistry and Physiology Division, University of Perugia, L. Severi Square, 06129 Perugia, Italy; lara.macchioni@unipg.it

**Keywords:** methylglyoxal, prostate cancer, bone metastases, osteoblasts, ROS, RAGE, NF-kB

## Abstract

Bone metastases from prostate cancer (PCa) result from a complex cross-talk between PCa cells and osteoblasts (OB). Thus, targeting this interplay has become an attractive strategy to interfere with PCa bone dissemination. The agents currently used in clinical trials have proved ineffective, boosting research to identify additional mechanisms that may be involved in this two-directional talk. Here, we investigated whether and how 5-hydro-5-methylimidazolone (MG-H1), a specific methylglyoxal (MG)-derived advanced glycation end product (AGE), was a novel player in the dialogue between PCa and OB to drive PCa bone metastases. Conditioned medium from osteotropic PC3 PCa cells, pre-treated or not with a specific MG scavenger, was administrated to human primary OB and cell morphology, mesenchymal trans-differentiation, pro-osteogenic determinants, PCa-specific molecules, and migration/invasion were studied by phase-contrast microscopy, real-time PCR, western blot and specific assays, respectively. We found that PC3 cells were able to release MG-H1 that, by binding to the receptor for AGEs (RAGE) on OB, reprogrammed them into a less-differentiate phenotype, endowed with some PCa-specific molecular features and malignant properties, in a mechanism involving reactive oxidative species (ROS) production and NF-kB pathway activation. These findings provide novel insights into the mechanisms of PCa osteoblastic metastases and foster in vivo research toward new therapeutic strategies interfering with PCa/OB cross-talk.

## 1. Introduction

Metastasis, responsible for most cancer deaths, is a complex and multi-step process during which primary cancer cells acquire migratory abilities, enter and survive in the bloodstream, and then reach, by extravasation, a secondary metastatic site where they start proliferating [1]. It is now widely accepted that a pivotal role in the onset of metastasis is played by the interaction between cancer cells and the stromal resident cells at the metastatic site [2]. According to the “seed and soil” theory by Paget, both cancer and tumor microenvironment (TME)-associated cells necessitate gaining novel and specific characteristics with the aim of creating suitable signaling networks to initiate a metastatic outgrowth [3]. An ideal “soil” for cancer cell metastasis is the bone. In fact, it represents a rich source of different chemokines, growth factors, and cytokines that facilitate cancer cell growth and colonization in addition to sustain their survival [4]. In particular, the cells of the bone participating in bone remodeling, namely osteoblasts (OB) and osteoclasts, are essential for cancer cell homing to and seeding in bone, cancer cell re-activation, macro-metastatic lesion growth, all these ultimately leading to extensive tumor cell colonization to bone [5].

Among the solid tumors that preferentially metastasize to the bone, there is prostate cancer (PCa). In this ambit, PCa is somewhat unique in its tendency to produce osteoblastic lesions [2,5]. Interestingly, these lesions are made of hypermineralized bone due to OB hyperactivity. However, since this process occurs abnormally, it results in a structure with a markedly reduced mechanical strength and, consequently, more susceptible to painful pathological fractures [6], which implies a diminished quality of life of the patients. In accordance with the general theory of Paget, also PCa cells are able to settle and grow in the skeleton due to cross-talk between the bone microenvironment and tumor cells [7]. In particular, bone provides chemotactic, adhesion, and growth factors that allow PCa cells to target and proliferate in the skeleton [8]. Reciprocally, PCa cells can produce factors that interfere with the physiological activity of bone cells and modulate bone remodeling [5], or even express OB-related genes, a condition known as “osteomimicry” [1,9]. Building on this evidence, targeting the cross-talk between PCa and bone cells has become an attractive strategy to interfere with PCa bone metastasis, and, indeed, several agents are currently in clinical trials to combat bone-metastatic PCa [5,6]. Unfortunately, these therapies have failed in showing prolonged median progression-free survival or overall survival [10,11,12]. Hence, focusing the research on understanding additional mechanisms and molecules through which PCa and bone cells realize their cross-talk to nurture the lethal phenotype of bone-metastatic PCa, is an urgent need. This would help developing novel and hopefully more effective therapeutic strategies with the ultimate aim to reduce morbidity and improve survival.

In addition, although considerable advancements have been made in recent years toward understanding metastatic PCa progression in bone, many questions still remain unsolved.

Methylglyoxal (MG) is a highly reactive by-product of glycolysis and a major precursor of advanced glycation end products (AGEs), especially 5-hydro-5-methylimidazolone (MG-H1) [13]. Prior evidence suggests that MG adducts, including MG-H1 [14], may signal through receptors for AGEs (RAGE) [15]. In general, engagement of RAGE by AGEs can activate transcription factor NF-kB, which increases reactive oxidative species (ROS) formation [14,16]. However, it has been recently demonstrated that the binding of AGEs with RAGE can also directly trigger ROS intracellular accumulation that in turn actives NF-kB pathway [17]. Of note, RAGE is expressed by OB [14,18], where it plays a pivotal role in bone metabolism [19].

In cells, MG is primarily metabolized by the glutathione-dependent enzyme glyoxalase 1 (Glo1). Hence, by modulating MG, Glo1 prevents intracellular accumulation of MG-H1 from participating as an upstream factor to MG-derived AGEs-mediated cell responses.

The role of the Glo1/MG-AGEs axis in PCa progression is today well defined [13]. Glo1 is overexpressed in advanced metastatic PCa, where, through specific MG-derived AGEs, it participates in cancer survival [20,21], PTEN/PI3K/AKT/mTOR signaling- and epithelial-to-mesenchymal transition (EMT)-related metastatic behavior [22,23], in addition, contributes to maintaining an immunosuppressive microenvironment through MG-H1-mediated PD-L1 up-regulation [24]. Whether Glo1/MG-H1 axis is also implicated in the mechanisms of PCa cells-driven bone metastases has never been investigated before.

Hence, recognized that bone metastases occurrence requires a constant dialogue between and reciprocal changes of both PCa and bone resident stromal cells, particularly OB [6], given that MG-AGEs, including MG-H1, can be secreted by cells [14] and act as paracrine signaling molecules through RAGE [15] to control bone metabolism [19], in the present study we investigated whether, and through which mechanism, MG-H1 might represent a novel agent implicated in the cross-talk between PCa and bone cells to possibly drive PCa osteotropic metastasis. Indeed, we found that metastatic PCa cells secrete MG-H1 that binds to RAGE on human primary OB to reprogram their phenotype into a less-differentiate one, endowed with some PCa-specific molecular features, in a mechanism involving ROS formation and NF-kB pathway activation.

## 2. Results

### 2.1. Conditioned Medium (CM) from PC3 Bone Metastasis-Derived PCa Cells Contains MG-Originated MG-H1

To investigate whether PC3 bone metastasis-derived PCa cells affected the homeostasis of human OB through MG-H1 secretion, PC3 were firstly cultured for 6 and 24 h in a humidified atmosphere at 37 °C, and the culture medium (conditioned medium, CM) was used to determine the levels of MG-H1 by a specific ELISA assay and Western blot (WB). As a control, the levels of MG-H1 were evaluated in the specific growth medium of OB, where they were cultured for the same period of time (control, CTR). Finally, the levels of MG-H1 were also determined in the fresh growth medium of both OB and PC3 cells before it was administrated to cells. As shown in Figure 1, MG-H1 was present only in PC3 CM, while nor CTR cells (Figure 1) nor OB and PC3 cells fresh medium (Appendix A) contained it. To prove that MG-H1 release was specifically associated with bone metastasis-derived PC3 cells, LNCaP cells, derived from a left supraclavicular lymph node metastasis and DU-145 cells, derived from a central nervous system metastasis, were grown for 24 h in RPMI medium [22] and MG-H1 was evaluated by WB in their CM. Finally, to show that MG-H1 was specifically released by PCa cells, also the non-cancerous PNT2 cells were cultured for 24 h in RPMI medium, and MG-H1 was measured in CM as above. As shown in Appendix A, none of these cells produced and secreted MG-H1, suggesting that MG-H1 release was specific to PCa cells disseminating to the bone.

### 2.2. Effect of CM from PC3 Cells on Human OB Viability and Morphology

We then treated human primary OB with PC3 cells CM, containing MG-H1, or with OB growth medium (CTR) for 6 and 24 h and investigated OB viability and morphology. As shown in Figure 2a, PC3 CM did not influence OB viability at both exposure times compared with CTR cells, suggesting that it was not toxic to OB. Conversely, it was able to modify OB morphology, evaluated by phase-contrast microscopy, at 24 h post-exposure (Figure 2b). In particular, OB exposed to CM assumed an elongated spindle-shaped morphology, very similar to that of mesenchymal cells, compared with the typical cobblestone, cuboidal one of the mature OB in CTR cells. Interestingly, PC3 CM exposure induced in OB also the formation of thin but well-developed filopodia-like protrusions (Figure 2b, arrows). Altogether, these results suggested that PC3 cells boosted OB into a mesenchymal-like phenotype and that, since OB originate from bone marrow mesenchymal cells, characterized by a spindle-shaped morphology and typically exhibiting multiple, thin protrusions [26], that PC3 cells were able to activate in OB a dedifferentiation program, very likely through MG-H1 secretion in a paracrine way.

### 2.3. CM from PC3 Cells Alters in OB the mRNA Expression of Specific Markers Associated with Mesenchymal Trans-Differentiation

To further prove that PC3 cells could induce the cellular regression of the mature phenotype of human OB to a less-differentiated, mesenchymal cell-like one, through the secretion of MG-H1, we evaluated, at the transcript level, the expression profile of markers typically associated with the mesenchymal trans-differentiation process, namely vimentin (VIM) [27], alpha-smooth muscle actin (α-SMA) [28], and transforming growth factor beta (TGF-β1), a pivotal factor in OB trans-differentiation [29]. In parallel, the transcript level of some OB-specific adhesion proteins, namely cadherin 11 (CDH11) [30], integrin β1 (ITGB1) [31], and integrin α3 (ITGA3) [32], were evaluated. In line with the morphological results, we found that CM from PC3 cells induced in OB a significant increase in VIM and α-SMA mesenchymal cells markers as well as of TGF-β1 (Figure 3a), while it decreased CDH11, ITGB1, and ITGA3 adhesion molecules (Figure 3b), compared with CTR cells. Hence, these data also confirmed at the molecular level the observed morphological changes, thus further supporting the hypothesis that PC3 cells were able to reprogram mature OB into a less-differentiated mesenchymal-like stage through MG-H1 release. Moreover, these findings suggested that the diminished levels of ITGB1 and ITGA3 expression possibly reflected an inefficient cell adhesion, typically associated with a mesenchymal phenotype.

### 2.4. CM from PC3 Cells Alters the mRNA Expression of Specific Markers Associated with OB Mature Phenotype

To further prove that PC3 cells were able to induce a shift of the mature phenotype of human OB to a less-differentiated one, we also determined the transcript level of some important molecules characteristic of OB mature phenotype, typically involved in osteogenic differentiation, namely runt-related transcription factor 2 (Runx2), collagen type I α1 (Col1α1), osteonectin (ON), osterix (OSX) [33] and the pre-osteoblast state marker, cluster of differentiation protein 44 (CD44) [34]. As shown in Figure 4, the mRNA expression of all OB-specific markers significantly decreased upon PC3 CM exposure at 24 h. Conversely, PC3 CM induced a significant increase in CD44 transcript levels with respect to OB grown in their own medium. Overall, this evidence further supported the role of PC3 cells in reverting back OB to a less-differentiated phenotype through MG-H1 release.

### 2.5. The Trans-Differentiation of OB Phenotype to A Mesenchymal-like One upon PC3 Cells CM Exposure Is Accompanied by Increased Migration/Invasion but Not Proliferation

As known, mesenchymal or mesenchymal-like cells are typically characterized by increased proliferation and motility [35]. Hence, we explored whether the trans-differentiation of OB phenotype to a mesenchymal-like one upon PC3 cells CM exposure was accompanied by increased migration/invasion and/or proliferation activities. As shown in Figure 5, OB exposed to PC3 cells CM for 24 h became more motile and invasive, while they did not change cell proliferation, thus further showing PC3 cells induced OB dedifferentiation also at the functional level, at least as far as migration and invasion are concerned, as expected for cells acquiring mesenchymal-like morphological traits.

### 2.6. CM from PC3 Cells Alters in OB the mRNA Expression of Proteins Involved in the Formation of Dynamic Cell Protrusions

To further support the cytoplasmic rearrangement processes observed by the morphological analysis and presumably associated with the observed increased OB motility, we then evaluated the mRNA levels of some key proteins involved in the cytoskeletal dynamic formation of cell protrusion (e.g., filipodia) [36]. In particular, we considered fascin (FASC), profilin 1 (PFN1), cofilin (COFN), and radixin (RADX) [32]. As shown in Figure 6, OB treated with PC3 CM presented a marked up-regulation of all the considered genes compared with control cells, also sustaining at the molecular level the occurrence of cytoplasmic rearrangement processes and, in particular, the formation of filopodia-like protrusions already evidenced by the morphological analysis.

### 2.7. CM from PC3 Cells Affects OB Mineralization Activity

PCa bone metastases are characterized by abnormally hypermineralized bone due to OB hyperactivity [6]. To investigate whether MG-H1-containing PC3 cells CM could somehow take part in this anomalous process, we first measured the mRNA expression of proteins typically involved in OB extracellular matrix mineralization, namely the non-collagenous protein osteocalcin (OC) and osteopontin (OP) [37]. In comparison with the control cells, the mRNA expression of OC and OP were significantly increased in OB exposed to PCa cells CM at 24 h post-exposure (Figure 7a). Subsequently, since OB mineralization also requires the deposition of the inorganic component (mineralized matrix) made up of hydroxyapatite crystals rich in calcium and phosphate [38], we evaluated the effect of CM from PC3 cells on OB mineral formation, evaluating calcium deposits by alizarin red staining method. As displayed in Figure 7b, PC3 cells-derived CM enhanced mineralization of OB at 24 h compared with controls.

### 2.8. CM from PC3 Cells Induces De Novo Expression of the Prostate-Specific Antigens PSA and PSMA in OB

Patients with metastatic PCa usually present high serum prostate-specific antigen (PSA) levels, and it has been suggested that PSA may participate in PCa osteoblastic metastases [39,40]. Hence, we investigated whether CM from PC3 cells containing MG-H1 was able to induce PSA and, possibly, membrane PSA (PSMA) expression, in OB. Surprisingly, we found that the mRNA expression of both prostate-specific markers was induced de novo in OB exposed to PC3 CM compared with control cells (Figure 8).

### 2.9. MG-derived MG-H1 Is a Novel Paracrine Factor Released by Bone Metastasis-Derived PC3 Cells to Reprogram Human OB into a Dedifferentiated, Mesenchymal-, and Malignant-like Phenotype

Overall, our results suggested that MG-H1, released from PC3 cancer cells, was able to induce OB phenotype regression consisting in the assumption of mesenchymal-like and prostate-specific traits and in gaining an enhanced mineralization capability, all these being hallmarks of PCa-associated bone metastases. To prove all this causatively, we pre-treated PC3 cells with aminoguanidine (AG), a specific MG scavenger [23], thus able to prevent MG-derived MG-H1 formation and release into the medium (Appendix A), for 6 h at 1 mM. Then, we collected the CM derived from PC3 treated with AG and administrated it to OB. Finally, OB morphology (Figure 9a), the expression of mesenchymal trans-differentiation-associated markers (VIM, α-SMA, TGF-β1, CDH11, ITGB1, ITGA3) (Figure 9b), the expression of markers associated with OB mature phenotype (Runx2, Col1α1, ON, OSX), the expression of the pre-osteoblast state marker CD44 (Figure 9c), migration and invasion (Figure 9d), the mRNA expression of proteins involved in the formation of filopodia-like protrusions (FASC, PFN1, COFN, RADX) (Figure 9e), bone mineralization through OC and OP expression and calcium deposits formation (Figure 10a), and the expression of the prostate-specific markers PSA/PSMA (Figure 10b), were evaluated compared with OB treated with PC3 cells CM not pre-treated with AG. As shown in Figure 9 and Figure 10, AG pre-treatment was able to completely abrogate the effects induced by MG-H1-containing CM on the above molecules and processes, thus confirming that MG-derived MG-H1 is a novel paracrine factor released by bone metastasis-derived PC3 cells to reprogram human OB into a mesenchymal-, malignant-like, and more invasive phenotype. These results were additionally confirmed by inducing MG-H1 accumulation through Glo1 silencing since this maneuver potentiated the studied responses (Appendix A).

### 2.10. MG-Derived MG-H1 Released by Bone Metastasis-Derived PC3 Cells Reprograms Human OB into a Mesenchymal-, Malignant-like Phenotype through a RAGE-Dependent Mechanism with the Involvement of ROS and NF-kB Signaling

MG-H1 can signal through receptor for AGEs (RAGE) [15] to trigger ROS intracellular accumulation that, in turn, actives NF-kB pathway [17]. Hence, we investigated whether MG-H1-induced OB dedifferentiation occurred through a RAGE-dependent mechanism, possibly involving reactive oxidative species (ROS) and the NF-kB pathway. Indeed, we found that PC3 CM-containing MG-H1 induced a significant increase in RAGE expression at both transcriptional and protein levels (Figure 11a), ROS intracellular amount (Figure 11b), and NF-kB-p65 nuclear levels (Figure 11c), compared with OB controls, while CM from PC3 pre-treated with AG prevented all these changes (Figure 11a,b), bringing NF-kB-p65 levels even under those of control (Figure 11c). These results suggested that MG-derived MG-H1 released by bone metastasis-derived PC3 cells reprograms human OB into a mesenchymal-, malignant-like phenotype through a RAGE-dependent mechanism. Importantly, upon MG-H1-containing PC3 CM exposure, blockade of RAGE with the high-affinity RAGE-specific inhibitor FPS-ZM1 [41,42], rescued OB dedifferentiation and re-wiring, evaluated by VIM, CADH11, Runx 2, CD44, migration, and PSMA levels (Figure 11d).

Overall, these findings indicated that MG-derived MG-H1 released by bone metastasis-derived PC3 cells reprogrammed human primary OB into a mesenchymal-, malignant-like phenotype through a RAGE-dependent mechanism with the involvement of ROS and NF-kB signaling.

### 2.11. CM from LNCaP Cells, Devoid of MG-H1, Does Not Reprogram OB into a Dedifferentiated Malignant-like Phenotype

The pivotal role played by MG-H1 in reprogramming OB into a dedifferentiated malignant-like phenotype was further confirmed by administrating CM from one of the cell lines devoid of MG-H1 (Appendix A), in particular LNCaP, on OB for 24 h. As shown in Appendix A, CM from LNCaP did not affect OB proliferation, morphology, expression of vimentin and CDH (specific markers associated with mesenchymal trans-differentiation), Runx2 and CD44 (specific markers associated with OB mature phenotype), migration/invasion, OC expression and calcium deposits (for extracellular matrix mineralization) as well as PSA/PSMA expression.

### 2.12. Circulating Levels of MG-H1 in Bone Metastasis-Bearing Patients

To provide a potential clinical value to our results, we finally measured the circulating levels of MG-H1 in patients with metastatic PCa (*n* = 30), previously enrolled [23]. In particular, among this cohort, we identified patients with bone metastases (*n* = 20) and patients with metastases to other organs (*n* = 10). In support of the mechanistic in vitro results, we found that patients with bone metastases presented significantly higher levels of MG-H1 than those with metastases to other organs (Figure 12).

## 3. Discussion

Prostate cancer (PCa) is the most common cancer in men in western countries, with a high incidence of bone metastases [43]. Due to osteoblasts (OB) hyperactivity, PCa bone metastases are typically characterized by an abnormal formation of the bone that is markedly more fragile and, consequently, more susceptible to fractures [6]. Colonization of PCa cells within the bone is complex and involves a network of interactions between cancer cells and OB [44] together with the production of paracrine signals and activation of signaling cascades modulated by numerous regulatory molecules. Hence, targeting the cross-talk between PCa and OB has become an attractive strategy to interfere with PCa bone metastasis. Indeed, several agents are currently in clinical trials to combat bone-metastatic PCa [5,6]. Unfortunately, therapies directed to PCa and/or OB-associated molecules have yielded disappointing clinical results [10,11,12], which suggests that our understanding of PCa/OB dialogue in this neoplasia is still limited. In this study, we describe, for the first time, that MG-H1 is a novel paracrine factor specifically released from osteotropic PCa cells to reprogram human primary OB into a less-differentiate, mesenchymal-like phenotype, characterized by increased motility and expressing some prostate-specific traits, in a mechanism involving RAGE-driven ROS production and NF-kB pathway activation (Figure 13).

In particular, the observed dedifferentiation switch was evident by the fact that OB exposed to MG-H1-containing CM assumed a spindle-shape morphology, reduced osteogenic differentiation markers, decreased adhesive proteins, enhanced mesenchymal-like components and cell protrusions, acquiring motility properties. Moreover, MG-H1 increased OB mineralization activity, which is a characteristic of PCa osteoblastic metastases. Interestingly, MG-H1 induced in OB de novo expression of PSA and PSMA prostate-specific molecules. Although apparently surprising, this observation indeed is in agreement with the role of PSA as a crucial player in preparing bone microenvironment to favor PCa osteoblastic metastases, acting as a protease against some matrix proteins [39]. Similarly, it has been demonstrated that also PSMA can have peptidase and hydrolase activities and be involved in tissue regeneration and repair [45]. Hence, its participation in the remodeling of the bone-metastatic niche to favor bone metastasis formation is very plausible. Moreover, this membrane protein, highly expressed in aggressive metastatic PCa, can also have a proangiogenic function, which implies a possible role for PSMA in the bone-metastatic microenvironment also in this scenario. Hence, altogether our findings suggest that MG-H1 is a novel player in the dialogue between PCa cells and OB to promote bone metastases. In particular, MG-H1, through the receptor RAGE, would reverse OB mature phenotype and impair some of OB specialized functions (motility and mineralization) to boost them to create an ad hoc niche to accommodate bone disseminating PCa cells and favor their growth during cancer progression. Hence, our results are in agreement and add novel insight into the “seed and soil” hypothesis by Paget, according to which, to facilitate the formation of metastases, the “soil” (metastatic microenvironment, OB in our case) must be able to nourish “seeds” (disseminated PCa cells). Ultimately, changes in the bone “soil” by PCa cells through MG-H1 secretion, as here observed, can create a favorable growth environment for the metastasis of PCa. We also demonstrated that MG-H1 is able to remodel the bone-metastatic niche through a mechanism involving RAGE-dependent ROS production and NF-kB pathway activation. It is known that NF-kB can trigger ROS formation [14,16]. Similarly, emerging evidence shows that ROS can activate the NF-kB pathway [17] so that a strong interplay exists between these two actors, which, for this, are not mutually exclusive. Hence, we would like to point out that once MG-H1/RAGE complex is formed and drives ROS and NF-kB production/activation, it is also possible that a concomitant amplification of the downstream effects on OB dedifferentiation might occur due to ROS/NF-kB-positive feed-forward loop within a precise regulatory circuitry. Another important pathway in the regulation of bone-remodeling processes is WNT/β-catenin signaling that has been shown to drive epithelial to mesenchymal transformation (EMT) of OB to induce their dedifferentiation [46]. Considering that here we observed a correlation between the assumption of mesenchymal traits with OB dedifferentiation and that oxidative stress, here induced by MG-H1, can activate the WNT pathway [47], the participation of this signaling pathway could also be very plausible, which deserves further investigation.

In the last decade, we have demonstrated that Glo1, by scavenging MG and, consequently by preventing MG-derived MG-H1 formation, plays a pivotal role in PCa progression through both apoptosis [20,21] and EMT [22,23] control. Very recently, we have also found that Glo1/MG-H1 axis contributes to maintaining an immunosuppressive microenvironment by modulating PD-L1 expression [24]. Here, we added further information to the oncogenic role of the Glo1/MG-H1 pathway in PCa, including it among the mechanisms of PCa cells/OB cross-talk essential to guide bone metastasis. More importantly, we provide a novel target to hinder or prevent bone dissemination, the major problem for PCa survival. Moreover, we would like to point out the umpteenth biological effect of MG-H1, namely that of driving de-differentiation, which was never reported before.

Finally, we found that patients with bone metastases presented significantly higher levels of MG-H1 than those with metastases to other organs. These in vivo results would support our mechanistic data and would seem to suggest a potential translational value of circulating MG-H1 as a bone metastases marker. Although further in vivo and epidemiologic studies are needed to prove this hypothesized translational value, our results would seem to head in this direction. Hence, in this regard, our results would open the way to additional studies aimed at investigating the putative role of MG-H1 circulating level as a biomarker of PCa osteotropism, useful to stratify PCa patient risk to develop bone metastases, which is fundamental to define personalized diagnostic and therapeutic strategies, possibly at the earliest stages of the disease. In fact, as mentioned above, the majority of patients with advanced PCa develop skeletal metastases, which may ultimately lead to serious complications, named “skeletal-related events” (SRE), that often dramatically impact on quality of life and survival.

## 4. Materials and Methods

### 4.1. Reagents

All of the chemicals used in the present study were analytical grade reagents from various sources. Phosphate-buffered saline (PBS), formaldehyde acetic acid and ammonium hydroxide, aminoguanidine bicarbonate (AG), MTT [3-(4,5-dimethylthiazol-2-yl)-2,5-diphenyltetrazolium bromide] and FPS-ZM1 were purchased from Merck Spa (Milan, Italy). Laemmli buffer and 2′–7′-dichlorofluorescein-diacetate (H_2_DCF-DA) were purchased by Thermo Fisher Scientific (Milan, Italy), Roti-Block from Prodotti Gianni (Milan, Italy), and bicinchoninic acid (BCA) kit from Thermo Fisher Scientific (Milan, Italy).

### 4.2. Cell Cultures

PC3, LNCaP, DU145, PNT2 cell lines were purchased by the American Type Culture Collection (ATCC) (Manassas, VA, USA) and cultured in RPMI medium supplemented with antimycotic and antibiotics (Thermo Fisher Scientific (Milan, Italy) at 37 °C and 5% CO_2_ [23,48]. Human primary osteoblasts (OB) were purchased from PromoCell (Heidelberg, Germany) and cultured in DMEM (Thermo Fisher Scientific (Milan, Italy) supplemented with 10% fetal calf serum (FCS, Thermo Fisher Scientific (Milan, Italy), antimycotic, and antibiotics (Thermo Fisher Scientific, Milan, Italy) [14].

### 4.3. Preparation of Conditioned Medium (CM) and Treatments

Cells were grown in 25 cm^2^ tissue culture flasks with their respective maintenance medium until they reached 90% confluence. Then, the medium was replaced with 10 mL of medium containing 0.5% charcoal-stripped FBS and continued to be cultured for 48 h. Supernatants were collected as CM. OB were cultured in CM for 6 and 24 h. OB cultured for the same period of time in their specific growth medium represented control cells (CTR). AG and FPS-ZM1 were prepared as stock solutions and diluted to the desired final concentrations immediately before use. The final concentrations of the compounds were as follows: AG (1 mM, for 6 h) and FPS-ZM1 (100 nM, 10 h). For DMSO-solubilized FPS-ZM1, the final DMSO concentration in incubations was 0.01%. Controls contained an identical volume of DMSO vehicles.

### 4.4. MG-H1 Detection

MG-H1 was measured either by the OxiSelectTM methylglyoxal competitive enzyme-linked immunosorbent assay (ELISA) kit (DBA Italia Srl, Segrate, Italy) according to the manufacturer’s instructions or by western blot using the anti-MG-H1 Ab (dilution 1:1000, Cell Biolabs, cat. # STA-011, DBA Italia Srl, Segrate, Italy).

### 4.5. Cell Lysis and Western Blot

Cells were lysed in radioimmunoprecipitation assay (RIPA) lysis buffer to extract total proteins [49,50]. Nuclear extracts were obtained using the FractionPREP Cell Fractionation kit (Biovision, Vinci-Biochem, Florence, Italy) according to the manufacturer’s instructions [49,50]. Western blot was performed with an equal protein concentration (40 μg) of samples in Laemmli buffer. After boiling them for 5 min, they were resolved on 10%, 12%, or 15% SDS-PAGE and blotted onto a nitrocellulose membrane with iBlot Dry Blotting System (Invitrogen, Milan, Italy). Non-specific binding sites were blocked in Roti-Block at room temperature for 1 h. Membranes were then incubated overnight at 4 °C with an appropriate dilution of the following primary specific Abs: mouse anti-MG-H1 mAb (dilution 1:1000, Cell Biolabs, cat. # STA-011, DBA Italia Srl, Segrate, Italy), mouse anti-RAGE (A-9) mAb (dilution 1:1000, Santa Cruz, cat. # sc-365154, DBA Italia Srl, Segrate, Italy), rabbit anti-NFkB p65 (D14E12) mAb (dilution 1:1000, cat. # 8242, Cell Signaling Technology, Milan, Italy), mouse anti-β-actin mAb (dilution 1:1000, Santa Cruz, cat. # sc- 47778, DBA Italia Srl, Segrate, Italy) and mouse anti-lamin B1 mAb (A-11) (dilution 1:1000, Santa Cruz, cat. # sc- 377000, DBA Italia Srl, Milan, Italy). Membranes were subsequently incubated with the appropriate HRP-conjugated secondary Ab at room temperature for 1 h to detect antigen-antibody complexes, and ECL was used as a revealing system (Amersham Pharmacia, Milan, Italy). The membranes were then appropriately re-probed with β-actin or lamin B1 as internal loading controls.

### 4.6. Cell Viability and Morphology

Cell viability was evaluated by MTT assay [14], while cell morphology by means of phase-contrast microscopy [14].

### 4.7. RNA Isolation, Reverse Transcription, and Real-Time Reverse Transcriptase-Polymerase Chain Reaction (RT-PCR) Analyses

TRIzol reagent (Thermo Fisher Scientific, Milan, Italy) was employed to isolate total cellular RNA. A total of 1 µg of RNA was used to synthesize cDNA using RevertAid™ H Minus First Strand cDNA Synthesis Kit (Thermo Fisher Scientific, Milan, Italy). Gene expression versus β-actin was evaluated by RT-PCR on an MX3000P Real-Time PCR System (Agilent Technology, Milan, Italy). The sequences of the oligonucleotide primers are reported in Table 1. PCR reactions were performed in a total volume of 20 µL with 25 ng of cDNA, 1X Brilliant II SYBR^®^ Green QPCR Master Mix, ROX Reference Dye, and 600 nM of specific primers. The thermal cycling conditions were 1 cycle at 95 °C for 5 min followed by 45 cycles at 95 °C for 20 s and 60 °C for 30 s. Melting curves were performed for all of the primer pairs in standard conditions. Comparative analysis of gene expression was performed by means of the 2−(∆∆*C*T) method [51,52].

### 4.8. Migration, Invasion and Cell Counting Assays

Migration was evaluated by the CytoSelect 24-Well Cell Migration Assay kit (cat. # CBA-100-5, DBA Italia S.r.l., Milan, Italy) and invasion by the CytoSelect 24-Well Cell Invasion Assay kit (cat. # CBA-110, DBA Italia S.r.l., Milan, Italy), according to the manufacturer’s instructions. Cell counting was performed using trypan blue (Thermo Fisher Scientific, Milan, Italy) [48].

### 4.9. Glo1-Specific Enzyme Activity

Glo1-specific enzyme activity was assessed as previously described [53]. Briefly, the assay solution contained 0.1 M sodium phosphate buffer (pH 7.2), 2 mM MG, and 1 mM GSH. The reaction was monitored spectrophotometrically by following the increase in absorbance at 240 nm and 25 °C. One unit of enzyme activity was defined as 1 μmol of S-d-lactoylglutathione produced per minute. To calculate Glo1-specific enzyme activity (s.a.), enzyme activity was related to the total protein concentration, determined by BCA.

### 4.10. Alizarin Red Staining

Alizarin red staining was performed with the Alizarin Red S Staining Quantification Assay (ARed-Q, cat. #867, ScienCell Research Laboratories, Rome, Italy) to evaluate calcium deposition. Briefly, cells were washed with PBS three times, fixed with 4% formaldehyde for 15 min at room temperature, washed with distilled water, and stained with 40 mM Alizarin red S for 30 min. Then, the dye was eluted using 10% acetic acid for 30 min at room temperature, neutralized with 10% ammonium hydroxide, and the absorbance was measured at 405 nm in a microplate reader.

### 4.11. Reactive Oxidative Species (ROS) Measurement

Assessment of ROS cellular levels, including levels of general reactive oxygen species and reactive nitrogen species, was performed as previously described [49,50], using the membrane-permeable 2′,7′-dichlorodihydrofluorescein diacetate (H_2_DCF-DA) fluorogenic dye.

### 4.12. Circulating Levels of MG-H1 in Samples from Metastasis-bearing Patients

OxiSelectTM Methylglyoxal Competitive ELISA kit (DBA Italia Srl, Segrate, Italy) was used to detect MG-H1 levels in lymphocytes from bone metastasis-bearing patients (*n* = 20) and patients with other metastases (*n* = 10). The study was conducted on residual material from patients previously enrolled for another study [54]. At the time of sampling (2005–2006), as part of the internal hospital protocol, oral informed consent from patients was routinely obtained. In this phase, among other clinical aspects, the patients were informed that part of the samples (blood samples) might have been used for research purposes without undermining the clinical/diagnostic analyses. Moreover, access to the data of patients and biological material had already been fully anonymized before the authors accessed them. Finally, observation of all rules concerning confidentiality and protection of personal data, in accordance with European Union, international, and national rules, was respected. The research has been carried out in accordance with the Declaration of Helsinki and guidelines of the Santa Maria of Misericordia Hospital of Perugia.

### 4.13. Statistical Analysis

Results were expressed as means ± standard deviation (SD) of three independent experiments. One-way analysis of variance with Dunnett’s correction was employed to determine differences among groups. Statistical significance was set at *p* < 0.05.

## 5. Conclusions

In summary, we demonstrated that the paracrine release of MG-H1 by osteotropic PCa cells reprograms OB, very likely to make them create a unique microenvironment to allow disseminating PCa cells to grow and flourish. Our data further improve our knowledge of the complex mechanisms underpinning PCa bone metastases onset and open the way to additional in vivo research aimed at studying novel therapeutic strategies, interfering with PCa/OB cross-talk in order to impair PCa metastases, improve patient quality of life, and, more important, reduce morbidity and increase survival. Of note, since bone is one of the most common metastatic sites for other malignancies [55,56], this research also sets the stage for additional studies aimed at investigating MG-H1 role in bone-metastatic responses and malignant behaviors in broader cancer contexts.

## Figures and Tables

**Figure 1 ijms-22-10191-f001:**
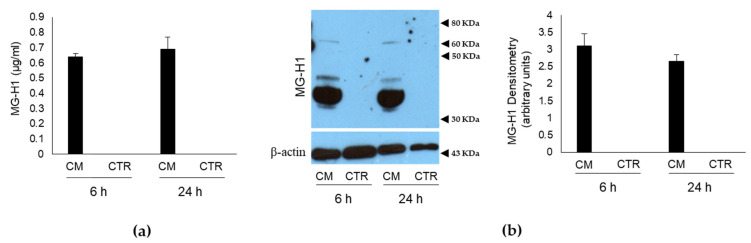
Level of methylglyoxal (MG)-derived 5-hydro-5-methylimidazolone (MG-H1) in the conditioned medium (CM) from PC3 bone metastasis-derived prostate cancer (PCa) cells. MG-H1 levels were measured by (**a**) a specific ELISA kit and (**b**) Western blot. In particular, PC3 were cultured for 6 and 24 h in a humidified atmosphere at 37 °C, and the culture medium they grew in (CM) was used to determine the levels of MG-H1. As a control (CTR), the levels of MG-H1 were also evaluated in the CM, where OB was cultured for the same period of time. β-actin was used as internal control, in agreement with Kwon et al. [25]. The histograms indicate the mean ± SD of three different cultures, and each was tested in duplicate.

**Figure 2 ijms-22-10191-f002:**
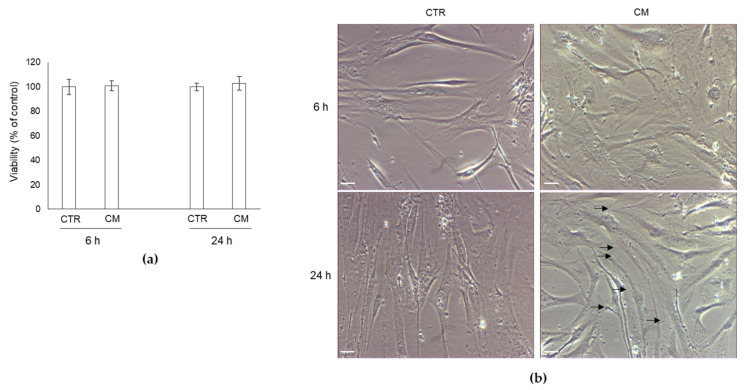
Effect of conditioned medium (CM) from PC3 bone metastasis-derived prostate cancer (PCa) cells on human primary osteoblasts (OB) viability and morphology. (**a**) cell viability measured by MTT assay and (**b**) cell morphology by means of light microscopy were evaluated in OB exposed for 6 and 24 h to CM from PC3 cells. Control cells (CTR) are represented by OB cultured for the same period of time in their specific growth medium. The histogram indicates the mean ± SD of three different cultures, and each was tested in duplicate. Scale bar = 30 μm. Arrows indicate filopodium-like protrusions.

**Figure 3 ijms-22-10191-f003:**
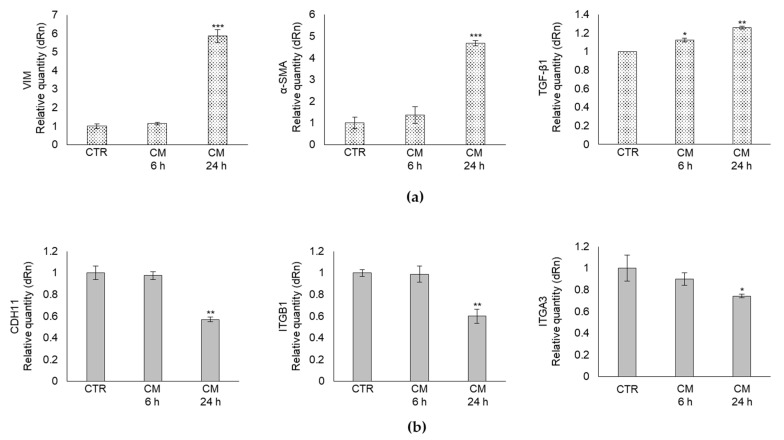
Effect of conditioned medium (CM) from PC3 bone metastasis-derived prostate cancer (PCa) cells on the mRNA expression of mesenchymal trans-differentiation-associated markers in human primary osteoblasts (OB). Transcript levels of (**a**) vimentin (VIM), alpha-smooth muscle actin (α-SMA) and transforming growth factor-β1 (TGF-β1) mesenchymal markers or (**b**), cadherin 11 (CDH11), integrin β1 (ITGB1) and integrin alpha 3 (ITGA3) adhesion molecules were evaluated by real-time PCR in OB exposed for 6 and 24 h to CM from PC3 cells. Control cells (CTR) are represented by OB cultured for the same period of time in their specific growth medium. Two experimental CTRs were performed for each exposure time; being both of them set at 1, each histogram reports only one CTR. The histogram indicates the mean ± SD of two different cultures, and each was tested in triplicate; * *p* < 0.05, ** *p* < 0.01, *** *p* < 0.001 versus CTR.

**Figure 4 ijms-22-10191-f004:**
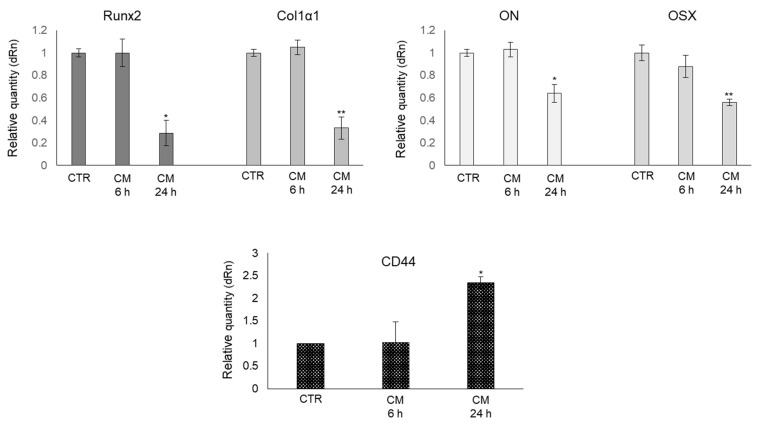
Effect of conditioned medium (CM) from PC3 bone metastasis-derived prostate cancer (PCa) cells on the mRNA expression of osteogenic differentiative markers runt-related transcription factor 2 (Runx2), collagen type I α1 (Col1α1), osteonectin (ON), osterix (OSX), and the pre-osteoblast state marker, cluster of differentiation protein 44 (CD44) in human primary osteoblasts (OB). Transcript levels were evaluated by real-time PCR in OB exposed for 6 and 24 h to CM from PC3 cells. Control cells (CTR) are represented by OB cultured for the same period of time in their specific growth medium. Two experimental CTRs were performed for each exposure time; being both of them set at 1, each histogram reports only one CTR. The histogram indicates the mean ± SD of two different cultures, and each was tested in triplicate. * *p* < 0.05, ** *p* < 0.01 versus CTR.

**Figure 5 ijms-22-10191-f005:**
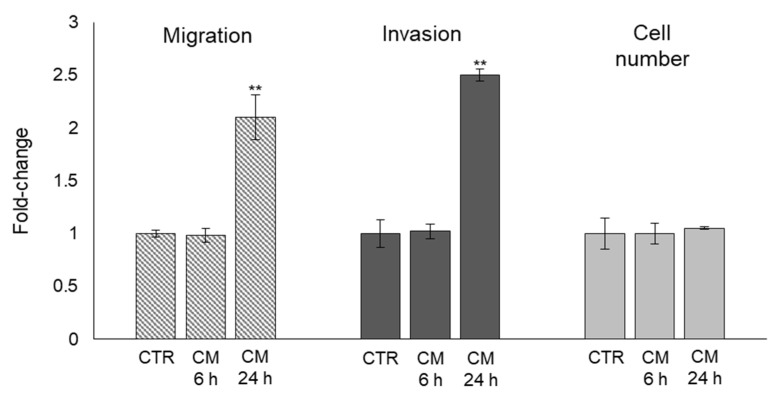
Effect of conditioned medium (CM) from PC3 bone metastasis-derived prostate cancer (PCa) cells on human primary osteoblasts (OB) migration, invasion, and proliferation. Migration and invasion were evaluated by specific assays, while cell proliferation by cell counting, in OB exposed for 6 and 24 h to CM from PC3 cells. Control cells (CTR) are represented by OB cultured for the same period of time in their specific growth medium. Two experimental CTRs were performed for each exposure time; being both of them set at 1, each histogram reports only one CTR. The histogram indicates the mean ± SD of two different cultures, and each was tested in triplicate. ** *p* < 0.01, versus CTR.

**Figure 6 ijms-22-10191-f006:**
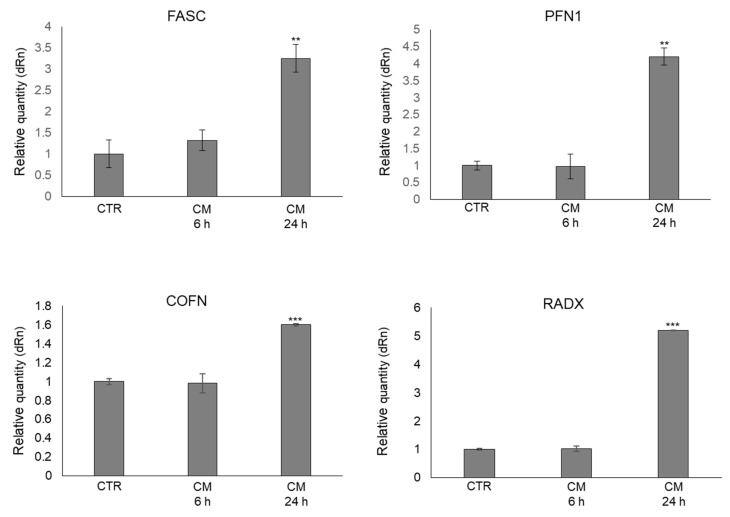
Effect of conditioned medium (CM) from PC3 bone metastasis-derived prostate cancer (PCa) cells on the mRNA expression of some key proteins involved in the cytoskeletal dynamic formation of cell protrusion in human primary osteoblasts (OB). Transcript levels were evaluated by real-time PCR in OB exposed for 6 and 24 h to CM from PC3 cells. Control cells (CTR) are represented by OB cultured for the same period of time in their specific growth medium. Two experimental CTRs were performed for each exposure time; being both of them set at 1, each histogram reports only one CTR. The histogram indicates the mean ± SD of two different cultures, and each was tested in triplicate. FASC, fascin; PFN1, profilin 1; COFN, cofilin; RADX, radixin. ** *p* < 0.01, *** *p* < 0.001, versus CTR.

**Figure 7 ijms-22-10191-f007:**
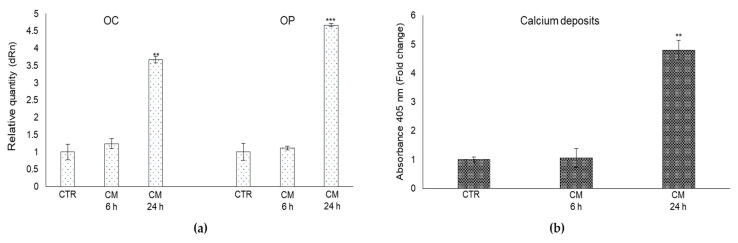
Effect of conditioned medium (CM) from PC3 bone metastasis-derived prostate cancer (PCa) cells on the mRNA expression of some key proteins typically involved in OB extracellular matrix mineralization, namely the non-collagenous protein osteocalcin (OC) and osteopontin (OP) (**a**), and on the formation of calcium deposits in human primary osteoblasts (OB) (**b**). Transcript levels were evaluated by real-time PCR while calcium deposits were measured by the alizarin red staining method in OB exposed for 6 and 24 h to CM from PC3 cells. Control cells (CTR) are represented by OB cultured for the same period of time in their specific growth medium. Two experimental CTRs were performed for each exposure time; being both of them set at 1, each histogram reports only one CTR. The histogram indicates the mean ± SD of two different cultures, and each was tested in triplicate. ** *p* < 0.01, *** *p* < 0.001.

**Figure 8 ijms-22-10191-f008:**
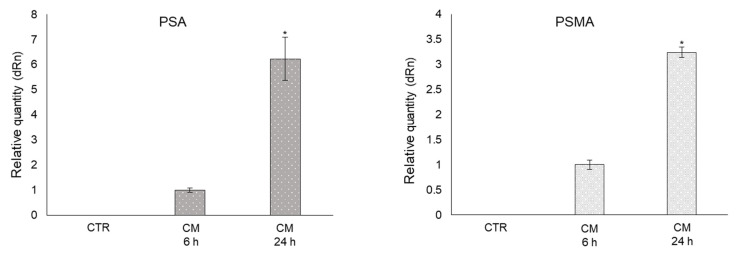
Conditioned medium (CM) from PC3 cells induces de novo expression of the prostate-specific antigens PSA and PSMA (prostate-specific membrane antigen) in OB. Transcript levels were evaluated by real-time PCR in OB exposed for 6 and 24 h to CM from PC3 cells. Control cells (CTR) are represented by OB cultured for the same period of time in their specific growth medium. Two experimental CTRs were performed for each exposure time; being both of them set at 1, each histogram reports only one CTR. The histogram indicates the mean ± SD of two different cultures, and each was tested in triplicate. * *p* < 0.05.

**Figure 9 ijms-22-10191-f009:**
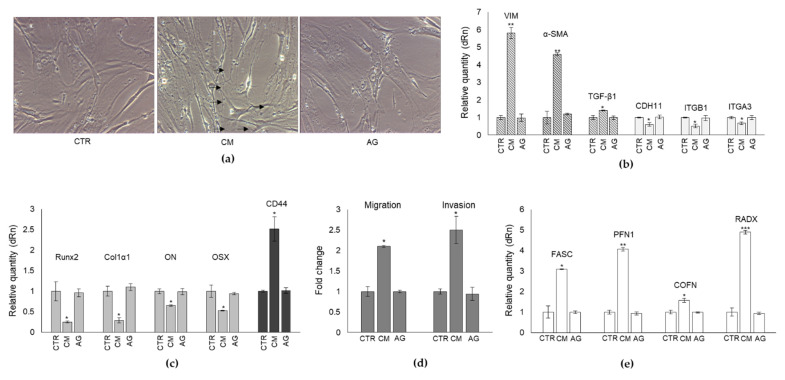
Methylglyoxal (MG)-derived 5-hydro-5-methylimidazolone (MG-H1) is a novel paracrine factor released by bone metastasis-derived PC3 cells to reprogram human osteoblasts (OB) into a mesenchymal-, malignant-like phenotype. (**a**) cell morphology by means of light microscopy, the expression of (**b**) mesenchymal trans-differentiation-associated markers (VIM, α-SMA, TGF-β1, CDH11, ITGB1, ITGA3), (**c**) markers associated with OB mature phenotype (Runx2, Col1α1, ON, OSX) and the pre-osteoblast state marker CD44 (**c**) and (**e**) proteins involved in the formation of filopodia-like protrusions (FASC, PFN1, COFN, RADX) as well as (**d**) migration and invasion were evaluated in OB exposed for 24 h to CM from PC3 cells and to CM from PC3 pre-treated (6 h) with the specific MG scavenger aminoguanidine (AG) (1 mM). Control cells (CTR) represent OB cultured for the same period of time in their specific growth medium. Transcript levels were evaluated by real-time PCR. The histogram indicates the mean ± SD of three different cultures, and each was tested in duplicate. Scale bar = 30 μm. Arrows indicate filopodium-like protrusions. VIM, vimentin; α-SMA, alpha smooth muscle actin; CDH11, cadherin 11; ITGB1, integrin β1; ITGA3, integrin α3; FASC, fascin; PFN1, profilin 1; COFN, cofilin; RADX, radixin. * *p* < 0.05, ** *p* < 0.01, *** *p* < 0.001, versus CTR and AG.

**Figure 10 ijms-22-10191-f010:**
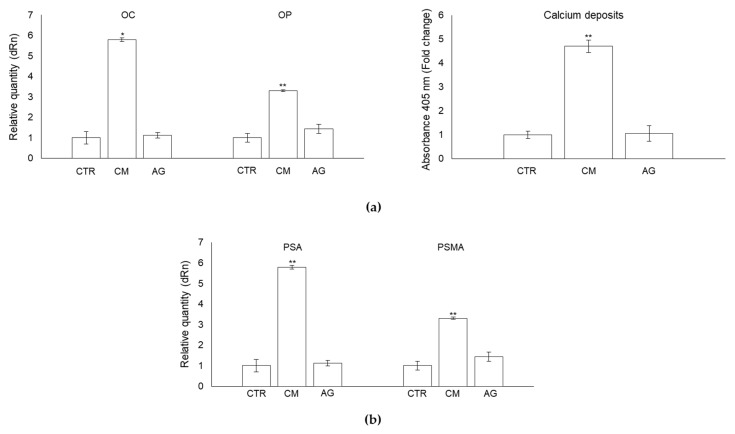
Methylglyoxal (MG)-derived 5-hydro-5-methylimidazolone (MG-H1) is a novel paracrine factor released by bone metastasis-derived PC3 cells to alter human osteoblasts (OB) mineralization and induce a prostate-specific phenotype. (**a**) expression of bone mineralization-participating proteins osteocalcin (OC) and osteopontin (OP) and calcium deposits formation, (**b**) mRNA expression of prostate-specific antigen (PSA) and prostate-specific membrane antigen (PSMA) evaluated in OB exposed for 24 h to conditioned medium (CM) from PC3 cells and to CM from PC3 pre-treated (6 h) with the specific MG scavenger aminoguanidine (AG) (1 mM). Control cells (CTR) represent OB cultured for the same period of time in their specific growth medium. Transcript levels were evaluated by real-time PCR while calcium deposits were measured by the alizarin red staining method. The histogram indicates the mean ± SD of three different cultures, and each was tested in duplicate. * *p* < 0.05, ** *p* < 0.01, versus CTR and AG.

**Figure 11 ijms-22-10191-f011:**
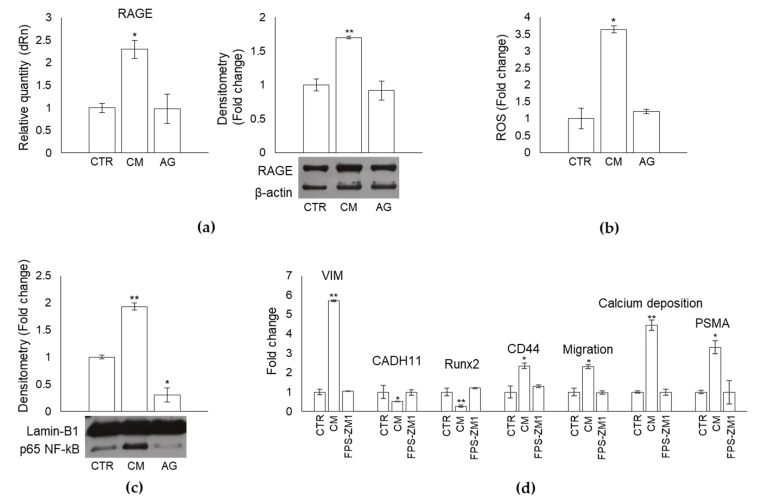
Methylglyoxal (MG)-derived 5-hydro-5-methylimidazolone (MG-H1) released by bone metastasis-derived PC3 cells reprograms human osteoblasts (OB) into a mesenchymal- malignant-like phenotype through a receptor for advanced glycation end products (RAGE)-dependent mechanism with the involvement of reactive oxidative species (ROS) and NF-kB signaling. (**a**) expression of RAGE, measured by real-time PCR and western blot; (**b**) intracellular levels of ROS, evaluated by 2’,7’-dichlorodihydrofluorescein diacetate; (**c**) p65 NF-kB nuclear expression by western blot and (**d**) trans-differentiation, migration, and mineralization-related responses in not exposed OB (CTR) or OB exposed for 24 h to conditioned medium (CM) from PC3 cells or to CM from PC3 pre-treated (6 h) with the specific MG scavenger aminoguanidine (AG) (1 mM) or to PC3 CM in the presence of the RAGE antagonist FPS-ZM1. Control cells (CTR) represent OB cultured for the same period of time in their specific growth medium. The histogram indicates the mean ± SD of three different cultures, and each was tested in duplicate. * *p* < 0.05, ** *p* < 0.01 versus CTR. VIM, vimentin; CADH11, cadherin 11; Runx2, runt-related transcription factor 2; CD44, cluster of differentiation protein 44; PSMA, prostate-specific membrane antigen.

**Figure 12 ijms-22-10191-f012:**
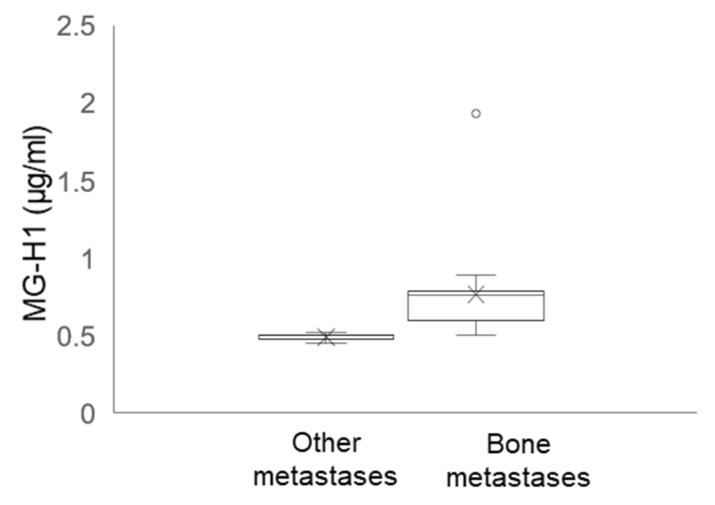
Circulating levels of 5-hydro-5-methylimidazolone (MG-H1) in bone metastasis-bearing patients divided into those with bone metastases (*n* = 20) and those with metastases from other organs (other metastases, *n* = 10).

**Figure 13 ijms-22-10191-f013:**
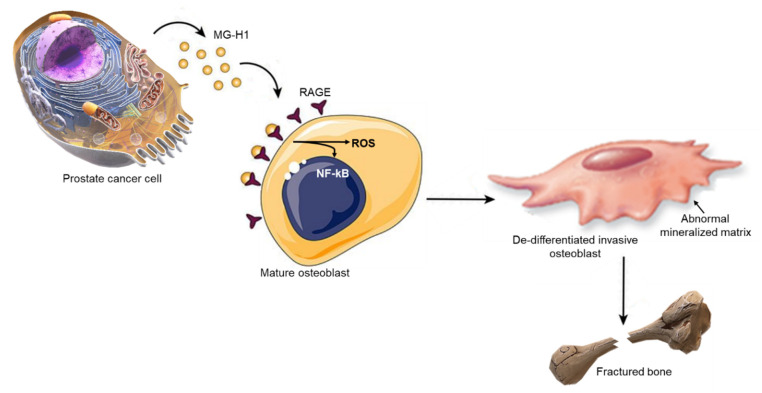
Metastatic prostate cancer (PCa) cells secrete methylglyoxal (MG)-derived 5-hydro-5-methylimidazolone (MG-H1) to reprogram human osteoblasts (OB) into a dedifferentiated, malignant-like phenotype: a possible novel player in prostate cancer bone metastases. MG-H1 released by PCa cells induces OB to dedifferentiate, assuming a spindle-shape morphology, reducing osteogenic differentiation markers, decreasing adhesive proteins, enhancing mesenchymal-like components and cell protrusions, acquiring motility properties. In addition, MG-H1 increased abnormal OB mineralization activity, leading to a bone structure with a markedly reduced mechanical strength and, consequently, more susceptible to pathological fractures. All these changes occurred via MG-H1 binding and activation of the receptor for advanced glycation end products (AGEs), RAGE. Hence, MG-H1 represents a novel player in the cross-talk between PCa cells and OB to create a favorable growth environment for bone metastases.

**Table 1 ijms-22-10191-t001:** Primers for qRT-PCR.

Gene	Forward Primer (5′-3′)	Reverse Primer (5′-3′)
VIM	GCACACAGCAAGGCGATGG	GGAGCGAGAGTGGCAGAGG
α-SMA	GGCATCATCACCAACTGGGACGAC	AGCACCGCCTGGATAGCCACATAC
TGF-β1	GGCGACCCACAGAGAGGAAATAG	AGGCAGAAATTGGCGTGGTAGC
CDH11	TGGCAGCAAGTATCCAATGG	TTTGGTTACGTGGTAGGCAC
ITGB1	TGATTGGCTGGAGGAATGTTA	GTTTCTGGACAAGGTGAGCAA
ITGA3	GGACCTTACAACGCCGAGTG	GGAGGCTCTTTGGCTTGTTTT
Runx2	GCTCTTCCCAAAGCCAGAGT	ATCCTGAC-GAAGTGCCAT
Col1α1	GAGGGCCAAGACGAAGACATC	CAGATCACGTCATCGCACAAC
ON	CCTGGAGACAAGGTGCTAACAT	CGAGTTCTCAGCCTGTGAGA
OSX	CCACCTACCCATCTGACT	GTTTGGCTCCACCACTCC
CD44	AGTCCCTGGATCACCGA	CCTCTTGGTTGCTGTCTCA
FASC	CTGGCTACACGCTGGAGTTC	CTGAGTCCCCTGCTGTCTCC
PFN1	TGGAGCAAACCCTACCCTT	AGCCCAGACACCGAACTTT
COFN	ATGCCCTCTATGATGCAACC	GCTTGATCCCTGTCAGCTTC
RADX	GAATCAGGAGCAGCTAGCAGCAGAACTT	TTGGTCTTTTCCAAGTCTTCCTGGGCTGCA
OC	TCACACTCCTCGCCCTATTGG	TCACACTCCTCGCCCTATTGG
OP	AGACCCCAAAAGTAAGGAAGAAG	GACAACCGTGGGAAAACAAATAAG
Glo1	CTCTCCAGAAAAGCTACACTTTGAG	CGAGGGTCTGAATTGCCATTG
β-actin	CACTCTTCCAGCCTTCCTTCC	ACAGCACTGTGTTGGCGTAC

## Data Availability

The data presented in this study are available on request from the corresponding author.

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
