# Peer review of "Metastatic Prostate Cancer Cells Secrete Methylglyoxal-Derived MG-H1 to Reprogram Human Osteoblasts into a Dedifferentiated, Malignant-like Phenotype: A Possible Novel Player in Prostate Cancer Bone Metastases"

_ijms, 2021, doi:10.3390/ijms221910191_

Round 1

Reviewer 1 Report

General comments

This paper described that MG-H1, a specific methylglyoxal (MG)-derived advanced glycation end product (AGE), is a novel player in the dialogue between prostate cancer and osteoblasts to drive prostate cancer bone metastases. There is a major concern that I would suggest.

Specific comments

Major

1)  All condition medium (CM) experiments used only one of cancer cell lines (PC3) and purchased osteoblasts. Even though LNCaP and DU145 did not express MG-H1, in order to emphasize that MG-H1 contribute to reprogram osteoblasts into a dedifferentiated malignant-like phenotype, the authors should show negative data (condition medium experiments using LNCaP or DU145).

Minor

1) In the result section, the authors mentioned about PNT2 cell line, but in the material and methods section, the authors mentioned PNT1 cell line. Which is correct?

Author Response

Reviewer #1

This paper described that MG-H1, a specific methylglyoxal (MG)-derived advanced glycation end product (AGE), is a novel player in the dialogue between prostate cancer and osteoblasts to drive prostate cancer bone metastases. There is a major concern that I would suggest.

Major

1)  All condition medium (CM) experiments used only one of cancer cell lines (PC3) and purchased osteoblasts. Even though LNCaP and DU145 did not express MG-H1, in order to emphasize that MG-H1 contribute to reprogram osteoblasts into a dedifferentiated malignant-like phenotype, the authors should show negative data (condition medium experiments using LNCaP or DU145).

Authors’ response: Authors thank the Reviewer very much for this comment. Actually, we had already performed some experiments where CM from LNCaP, devoid of MG-H1 (Figure S1), was administrated to osteoblasts for 24 hours and evaluated osteoblasts viability and morphology, together with the expression of vimentin and CDH (as specific markers associated with mesenchymal trans-differentiation), Runx2 and CD44 (as specific markers associated with OB mature phenotype), migration/invasion, OC expression and calcium deposits (for extracellular matrix mineralization) as well as PSA/PSMA expression. Since exposure of osteoblasts to LNCaP CM did not affect all these biological responses, we did not show these negative results but just the levels of MG-H1, the upstream player in the cross-talk between prostate cancer cells and osteoblasts. Anyway, the Reviewer is absolutely right: showing these negative results further helps in strengthening our conclusions about the role of MG-H1 in contributing to reprogram osteoblasts into a dedifferentiated malignant-like phenotype. Hence, we have now added these results in a new paragraph (please, see page 12, rows 352-361) and show them in Figure S4.

Minor

1) In the result section, the authors mentioned about PNT2 cell line, but in the material and methods section, the authors mentioned PNT1 cell line. Which is correct?

Authors’ response: PNT2 is the correct name of the cell line we worked with. Authors apologize for this oversight during typing. We have now corrected in M&M (pag 16, row 479) and changed PNT1 to PNT2.

Reviewer 2 Report

Review ijms-1371128: Metastatic prostate cancer cells secrete methylglyoxal-derived MG-H1 to reprogram human osteoblasts into a dedifferentiated, malignant-like phenotype: a possible novel player in prostate cancer bone metastases

The paper presents the results of in vitro studies on the role of MG-H1, a specific advanced glycation end product (AGE) derived from methylglyoxal (MG) in the cellular chain of dependence in the process of bone metastasis formation in prostate cancer.

The obtained results constitute a new contribution to the understanding of the mechanism of bone metastases formation not only in prostate cancer, but also in other neoplasms with a predilection for osteoblasts.

The manuscript is well written and clear, the quantity and quality of the work is commendable, and the proposed approach may be of use to the research community for similar research.

The test protocol was designed correctly with all aspects to exclude methodological errors.

I recommend the manuscript for publication, with only one suggestion to change the layout of the manuscript:

 - Moving the Materials and Methods section before the results part - keeping the classic layout will allow readers to fully understand the authors' approach to the topic

Author Response

Reviewer #2

The paper presents the results of in vitro studies on the role of MG-H1, a specific advanced glycation end product (AGE) derived from methylglyoxal (MG) in the cellular chain of dependence in the process of bone metastasis formation in prostate cancer.

The obtained results constitute a new contribution to the understanding of the mechanism of bone metastases formation not only in prostate cancer, but also in other neoplasms with a predilection for osteoblasts.

The manuscript is well written and clear, the quantity and quality of the work is commendable, and the proposed approach may be of use to the research community for similar research.

The test protocol was designed correctly with all aspects to exclude methodological errors.

Authors’ response: Authors thank the Reviewer very much for his/her appreciation on our study.

I recommend the manuscript for publication, with only one suggestion to change the layout of the manuscript:

 - Moving the Materials and Methods section before the results part - keeping the classic layout will allow readers to fully understand the authors' approach to the topic

Authors’ response: Authors agree with the Reviewer. However, the journal requires that the “Materials and Methods” section follows the “Results” section. A template is available with this layout in the homepage of the journal.

Reviewer 3 Report

Congratulations for this excellent article

Author Response

 Reviewer #3

Congratulations for this excellent article

Authors’ response: Authors thank the Reviewer very much for his/her appreciation on our study.

Round 2

Reviewer 1 Report

Accept in present form